The mortality of hospitalized patients with COVID-19 and non-cirrhotic chronic liver disease: a retrospective multi-center study

Wu Pei-Jui 1
Feng I-Che 1
Lai Chih-Cheng 2 3
http://orcid.org/0000-0001-5925-8477 Ho Chung-Han 3
Kan Wei-Chih 4
Sheu Ming-Jen 1 hmj@mail.chimei.org.tw
Kuo Hsing-Tao 1 kht@mail.chimei.org.tw
1 Division of Hepatogastroenterology, Department of Internal Medicine, Chi Mei Medical Center , Tainan , Taiwan
2 Department of Hospital Medicine, Chi Mei Medical Center , Tainan , Taiwan
3 Department of Medical Research, Chi Mei Medical Center , Tainan , Taiwan
4 Department of Internal Medicine, Chi Mei Medical Center , Tainan , Taiwan
Adegboye Oyelola
Electronic publication date: 2023 Dec 4
Publication date: 2023
Volume: 11
Electronic Location ID: e16582
Received 2023 Jun 20; Accepted 2023 Nov 13
Copyright: © 2023 Wu et al.
Copyright year: 2023
Copyright holder: Wu et al.
License: This is an open access article distributed under the terms of the Creative Commons Attribution License, which permits unrestricted use, distribution, reproduction and adaptation in any medium and for any purpose provided that it is properly attributed. For attribution, the original author(s), title, publication source (PeerJ) and either DOI or URL of the article must be cited.
License URL: https://creativecommons.org/licenses/by/4.0/

Keywords: ALBI, APRI, CLD, COVID-19, FIB-4, Non-cirrhotic chronic liver disease, Omicron, SARS-CoV-2, Mortality predictor, Hospitalization

Funding: The authors received no funding for this work.

==============================
Background

Patients with chronic liver disease (CLD) have a higher risk of mortality when infected with severe acute respiratory syndrome coronavirus 2. Although the fibrosis-4 (FIB-4) index, aspartate aminotransferase-to-platelet ratio index (APRI), and albumin-bilirubin grade (ALBI) score can predict mortality in CLD, their correlation with the clinical outcomes of CLD patients with coronavirus disease 2019 (COVID-19) is unclear. This study aimed to investigate the association between the liver severity and the mortality in hospitalized patients with non-cirrhotic CLD and COVID-19.

Methods

This retrospective study analyzed 231 patients with non-cirrhotic CLD and COVID-19. Clinical characteristics, laboratory data, including liver status indices, and clinical outcomes were assessed to determine the correlation between liver status indices and the mortality among patients with non-cirrhotic CLD and COVID-19.

Results

Non-survivors had higher levels of prothrombin time-international normalized ratio (PT-INR), alanine aminotransferase, aspartate aminotransferase, and high-sensitivity C-reactive protein (hs-CRP) and lower albumin levels. Multivariable analysis showed that ALBI grade 3 (odds ratio (OR): 22.80, 95% confidence interval (CI) [1.70–305.38], p = 0.018), FIB-4 index ≥ 3.25 (OR: 10.62, 95% CI [1.12–100.31], p = 0.039), PT-INR (OR: 19.81, 95% CI [1.31–299.49], p = 0.031), hs-CRP (OR: 1.02, 95% CI [1.01–1.02], p = 0.001), albumin level (OR: 0.08, 95% CI [0.02–0.39], p = 0.002), and use of vasopressors (OR: 4.98, 95% CI [1.27–19.46], p = 0.021) were associated with the mortality.

Conclusion

The ALBI grade 3 and FIB-4 index ≥ 3.25, higher PT-INR, hsCRP levels and lower albumin levels could be associated with mortality in non-cirrhotic CLD patients with COVID-19. Clinicians could assess the ALBI grade, FIB-4 index, PT-INR, hs-CRP, and albumin levels of patients with non-cirrhotic CLD upon admission.

Introduction

In December 2019, coronavirus disease 2019 (COVID-19) was first identified in Wuhan, China, and its rapid spread led to a pandemic (Lai et al., 2020). COVID-19 is transmitted through droplets or direct contact and is caused by severe acute respiratory syndrome coronavirus 2 (SARS-CoV-2). The virus is highly transmissible and prone to frequent mutations due to its ribonucleic acid nature. Several vaccines have been developed and have been shown to reduce complications and mortality rates, but they do not offer complete protection against the infection. As of September 28, 2023, 6,958,499 deaths have been reported by the World Health Organization.

COVID-19 has a worse prognosis in patients with comorbidities due to immune dysfunction and coagulopathy (Marjot et al., 2021b). Chronic liver disease (CLD) is a common comorbidity among patients with COVID-19. Patients with pre-existing CLD are more vulnerable to COVID-19 and have a higher risk of developing acute-on-chronic liver failure (ACLF) than healthy individuals (Oyelade, Alqahtani & Canciani, 2020; Shiri Aghbash et al., 2022). COVID-19 comorbid with CLD may lead to mortality, predominantly due to respiratory failure, followed by ACLF (Oyelade, Alqahtani & Canciani, 2020; Marjot et al., 2021a; Sarin et al., 2020).

Early recognition of high-risk individuals, particularly those with comorbidities such as non-cirrhotic CLD, is important for preventing mortality. In addition to basic laboratory data, Child-Pugh class and Model for End-stage Liver Disease scores are commonly used to evaluate disease severity and predict mortality in cirrhosis. However, a decisive index to estimate outcomes of non-cirrhotic CLD is lacking. The fibrosis-4 (FIB-4) index (Vallet-Pichard et al., 2007) and aspartate aminotransferase-to-platelet ratio index (APRI) (Wai et al., 2003) are simple approaches to assessing fibrosis in cirrhotic and non-cirrhotic CLD. Additionally, the novel albumin-bilirubin grade (ALBI) score has recently been used to assess the prognosis of all types and stages of CLD, not just hepatocellular carcinoma (Toyoda & Johnson, 2022; Demirtas et al., 2021).

However, most articles (Marjot et al., 2021b; Oyelade, Alqahtani & Canciani, 2020; Shiri Aghbash et al., 2022; Marjot et al., 2021a; Sarin et al., 2020) emphasized patients with cirrhosis rather than those with non-cirrhotic CLD. Therefore, we addressed the group of non-cirrhotic CLD patients and aimed to analyze whether the clinical features, laboratory data, and aforementioned scores could be correlated with mortality in hospitalized patients with COVID-19.

Materials and Methods

Study design and patients

This study retrospectively enrolled all hospitalized patients diagnosed with the omicron variant of COVID-19 comorbid with CLD between January 1, 2022, and September 30, 2022, at Chi Mei Medical Center and two affiliated hospitals; patient’s data were obtained from the electronic medical record databases of the hospitals. Basic characteristics, laboratory data, treatment, clinical course, severity, and liver status indices were collected for analysis.

Patients who met any of the following criteria were excluded: (a) age < 20 years old; (b) no evidence of CLD on ultrasonography, computed tomography, or magnetic resonance imaging; (c) fatty liver disease (FLD) without metabolic dysfunction (≤2 metabolic comorbidities); (d) unavailability of data, including the levels of albumin, total bilirubin, alanine aminotransferase (ALT), and aspartate aminotransferase (AST), prothrombin time-international normalized ratio (PT-INR), height, weight, FIB-4 index, APRI, and ALBI score. (e) Lack of clinical data on ascites or hepatic encephalopathy.

The study was approved by the Institutional Review Board of Chimei Medical Center (IRB Serial No.:11111-007). The informed consent requirement was waived due to the study’s retrospective nature.

Outcomes

We compared basic characteristics, laboratory data, treatment, clinical course, severity, and indices of liver status between survivors and non-survivors. The primary outcome of this study was to determine whether liver status could be associated with the mortality in patients with non-cirrhotic CLD who contracted the SARS-CoV-2 infection. The secondary outcome was to identify the risk factors related to all-cause in-hospital mortality among patients with non-cirrhotic CLD.

Definitions and variables

Chronic liver disease (CLD): Liver dysfunction lasting more than 6 months (Sharma & Nagalli, 2022). CLD was diagnosed in patients based on clinical features (history and laboratory data), imaging studies (ultrasound, computed tomography, magnetic resonance imaging), or liver biopsy, including cirrhotic and non-cirrhotic CLD.

Cirrhosis: Cirrhosis is diffuse hepatic fibrosis with the replacement of normal liver parenchyma by nodules after repeatedly destruction and regeneration (Sharma & John, 2022). Cirrhosis was diagnosed in patients based on clinical features, imaging/endoscopic studies, or liver biopsy.

Non-cirrhotic chronic liver disease (non-cirrhotic CLD): Non-cirrhotic CLD was diagnosed in patients who met the diagnostic criteria for CLD but had no evidence of cirrhosis.

Metabolic-associated fatty liver disease (MAFLD): Based on the diagnostic criteria of MAFLD (Lin et al., 2020), our patients were diagnosed with MAFLD if they had evidence of FLD on the imaging study and met at least one of the following criteria: (a) body mass index (BMI) >23 kg/m2; (b) history of type two diabetes mellitus; (c) ≥2 metabolic dysregulations, including hypertension, hypertriglyceridemia, and elevated high-density lipoprotein levels.

Acute liver injury (ALI): ALI was defined as having either ALT or AST ≥ 2 folds of the normal upper limit (ULN) within 4 weeks with previously normal liver enzymes.

Acute-on-chronic liver failure (ACLF): we modified the definition from Asian Pacific Association for the Study of the Liver consensus of ACLF (Sarin et al., 2014), and provisions of National Health Insurance of Taiwan (National Health Insurance Administration of Taiwan, 2023). ACLF was defined as the presence of any of the following within 4 weeks: (a) total bilirubin level ≥ 2 mg/dL, (b) PT-INR ≥ 1.5, (c) hepatic encephalopathy, (d) uncontrolled ascites, or (e) variceal bleeding.

Albumin-bilirubin grade (ALBI) score: ALBI score = (log10 bilirubin × 0.66) + (albumin × −0.085), where bilirubin is in μmol/L and albumin in g/L. ALBI score graded as grade 1 (ALBI score ≤ −2.60), grade 2 (score > −2.60 to ≤ −1.39), and grade 3 (score > −1.39) (Toyoda & Johnson, 2022; Demirtas et al., 2021).

Fibrosis-4 (FIB-4) index: FIB-4 = [age × AST (U/L)]/[platelet count (103/µL) × √ALT (U/L)]. FIB-4 index ≥ 3.25 indicates advanced fibrosis, equivalent to a METAVIR fibrosis score ≥ F3. FIB-4 index between 1.45 to 3.25 is roughly equivalent to a METAVIR fibrosis score F1–F2. FIB-4 index < 1.45 indicated no or mild fibrosis, equivalent to a METAVIR fibrosis score F0–F1 (Vallet-Pichard et al., 2007).

Aspartate aminotransferase-to-platelet ratio index (APRI): APRI = [AST (U/L) ÷ AST <ULN> (U/L)] × 100/platelet count (103/µL). APRI ≥ 1.5 is equivalent to advanced fibrosis. APRI between 0.5 to 1.5 indicated intermediate fibrosis. APRI < 0.5 indicated no or mild fibrosis (Wai et al., 2003).

Statistical analysis

Categorical variables were presented as frequencies with percentages, and the difference between the survivor and non-survivor groups was compared using the χ2 test or Fisher’s exact test. Continuous variables were computed as medians with interquartile ranges, and the differences between the survivor and non-survivor groups were compared using the Wilcoxon rank-sum test according to the distribution. The association between risk factors and all-cause mortality was calculated using logistic regression to estimate the odds ratios (ORs) with 95% confidence intervals (CIs). Crude and adjusted ORs were presented. To construct the multivariable logistic regression model and present adjusted ORs, we selected variables based on the criteria of crude ORs with p-values < 0.05. All analyses were performed using SAS version 9.4 (SAS Institute, Cary, NC, USA). Statistical significance was set at p < 0.05.

Results

Clinical characteristics

A total of 616 patients with CLD, including 484 patients with non-cirrhotic CLD and 132 with cirrhosis, were identified from 3,087 hospitalized patients who tested positive for COVID-19 using real-time reverse transcription-polymerase chain reaction. Of them, 324 were excluded based on the exclusion criteria. After excluding 61 patients with cirrhosis, 231 patients with non-cirrhotic CLD were included in our study (Fig. 1). These patients were classified into four subgroups based on the etiology of CLD: MAFLD (47.19%), alcoholic (19.05%), viral (30.74%), and others (3.02%).

Figure 1 Patient enrollment flow diagram.

The mortality rate of all 3,087 patients was 10.92% (n = 337). The all-cause mortality rates of patients with non-cirrhotic CLD and cirrhosis were 11.26% and 16.39%, respectively. The baseline characteristics of the survivor and non-survivor groups of hospitalized COVID-19 patients with non-cirrhotic CLD are summarized in Table 1. The mean age of the patients was 68.28 years, with male predominance (148 males, 64.07%), and 60.17% of them had received at least one dose of the vaccine. The median age distribution (68.00 years (60–78) vs. 68.50 years (60.50–81.80), p = 0.592) was not significantly different between the survivor and non-survivor groups, so was the distribution of sex (p = 0.472), etiologies of CLD (p = 0.192), BMI (p = 0.118), and comorbidities. Additionally, non-survivors had longer hospital stays (p = 0.010), higher rates of intensive care unit (ICU) admission (p = 0.049), and increased use of vasopressors (p < 0.001) and ventilators (p = 0.037).

Table 1 Characteristics of patients with non-cirrhotic CLD following COVID-19.

	Survivor (N = 205)	Non-survivor (N = 26)	p-value	
Median (Q1–Q3)/n (%)	Median (Q1–Q3)/n (%)	
Characteristics				
Age (years)	68.00 (61.00–78.00)	68.50 (61.00–81.00)	0.592	
Sex			0.472	
Male	133 (64.88)	15 (57.69)		
Female	72 (35.12)	11 (42.31)		
BMI (kg/m2)	23.85 (21.05–26.68)	22.74 (19.68–25.22)	0.118	
Vaccinated times			0.261	
0	79 (38.54)	13 (50.00)		
≥1	126 (61.46)	13 (50.00)		
Etiology			0.192	
MAFLD	100 (48.78)	9 (34.62)		
Alcohol	37 (18.05)	7 (26.92)		
Virus‡	63 (30.73)	8 (30.77)		
HBV	46 (22.44)	7 (26.92)	0.609	
HCV	26 (12.68)	1 (3.85)	0.328	
Others	5 (2.44)	2 (7.69)		
Medical history				
Cardiovascular	83 (40.49)	12 (46.15)	0.580	
Chronic kidney disease	40 (19.51)	8 (30.77)	0.183	
Diabetes	84 (40.98)	14 (53.85)	0.211	
Hyperlipidemia	31 (15.12)	3 (11.54)	0.446	
Malignancy	87 (42.44)	11 (42.31)	0.990	
HCC	15 (7.32)	1 (3.85)	1.000	
Disease course and severity				
Length of hospital (days)	12.00 (8.00–23.00)	24.50 (15.00–29.00)	0.010	
ICU admission	30 (14.63)	8 (30.77)	0.049	
The use of vasopressors	24 (11.71)	13 (50.00)	<0.001	
The use of ventilator	18 (8.78)	6 (23.08)	0.037	
Notes:

Abbreviations: BMI, body mass index; CLD, chronic liver disease; HBV, hepatitis B virus; HCV, hepatitis C virus; ICU, intensive care unit; MAFLD, metabolic-associated fatty liver disease.

‡Several patients had concomitant HBV and HCV infection; as a result, the total number of HBV plus HCV exceed those of virus.

Bold values indicate p < 0.05.

Laboratory data

Laboratory data and indices are summarized in Table 2. Compared to survivors, non-survivors had significantly higher PT-INR (p < 0.001), ALT (p = 0.033), AST (p = 0.005), and high-sensitivity C-reactive protein (hs-CRP) (p < 0.001) levels, and lower albumin levels (p < 0.001). Additionally, a significant difference was found between survivors and non-survivors regarding ALI (p = 0.003) and ACLF (p < 0.001). Furthermore, the liver status and fibrosis indices were significantly different between the two groups stratified according to the ALBI score (p < 0.001) and FIB-4 index (p = 0.004). In contrast, no significant difference was found between the groups stratified according to the APRI score (p = 0.078). The groups had no significant differences in total bilirubin (p = 0.544) or platelet levels (p = 0.932).

Table 2 Laboratory data and indexes at admission in patients with non-cirrhotic CLD following COVID-19 infection.

	Survivor (N = 205)	Non-survivor (N = 26)	p-value	
Median (Q1–Q3)/n (%)	Median (Q1–Q3)/n (%)	
Laboratory data				
Platelet (109/L)	180.00 (134.00–250.00)	171.50 (144.00–255.00)	0.932	
PT-INR	1.16 (1.07–1.23)	1.32 (1.23–1.40)	<0.001	
Total bilirubin (mg/dL)	0.60 (0.41–0.90)	0.65 (0.46–0.90)	0.544	
ALT (U/L)	20.00 (15.00–37.00)	34.00 (20.00–76.00)	0.033	
AST (U/L)	27.00 (19.00–45.00)	57.00 (27.00–95.00)	0.005	
Albumin (g/dL)	3.30 (3.10–3.70)	2.75 (2.40–3.20)	<0.001	
hs-CRP (mg/L)	23.35 (5.9–78.90)	121.15 (41.67–226.70)	<0.001	
COVID-19–related liver injury				
ACLF or decompensation	11 (5.37)	8 (30.77)	0.001	
Acute liver injury	33 (16.10)	11 (42.31)	0.003	
ALBI score			<0.001	
Grade 1	46 (22.44)	1 (3.85)		
Grade 2	149 (72.68)	18 (69.23)		
Grade 3	10 (4.88)	7 (26.92)		
FIB-4 index			0.004	
<1.45	108 (52.68)	7 (26.92)		
1.45–3.25	72 (35.12)	10 (38.46)		
≥3.25	25 (12.20)	9 (34.62)		
APRI			0.078	
<0.5	104 (50.73)	8 (30.77)		
0.5–1.5	69 (33.66)	10 (38.46)		
≥1.5	32 (15.61)	8 (30.77)		
Note:

Abbreviations: ACLF, acute-on-chronic liver failure; APRI, aspartate aminotransferase to platelet ratio index; AST, aspartate aminotransferase; ALT, alanine aminotransferase; ALBI, albumin-bilirubin grade; CLD, chronic liver disease; COVID-19, coronavirus disease 2019; FIB-4, fibrosis-4; hs-CRP, high-sensitivity C-reactive protein; PT-INR, prothrombin time-international normalized ratio.

Bold values indicate p < 0.05.

Analysis of risk factors and mortality between survivor and non-survivor groups

We analyzed the association using logistic regression, and the results are presented in Table 3. Univariate logistic regression analysis showed significant differences between non-survivors and survivors in PT-INR (p = 0.001), hs-CRP level (p < 0.001), albumin level (p < 0.001), vasopressor use (p < 0.001), mechanical ventilation use (p = 0.031), ALI (p = 0.002), and ACLF (p < 0.001). Multivariable logistic regression analysis showed a significant difference in PT-INR (OR: 19.81, 95% CI [1.31–299.49], p = 0.031), hs-CRP level (OR: 1.02, 95% CI [1.01–1.02], p = 0.001), albumin level (OR: 0.08, 95% CI: 0.02–0.39, p = 0.002), and the use of vasopressors (OR: 4.98, 95% CI [1.27–19.46], p = 0.021) between the survivor and non-survivor groups after adjusting for confounding factors.

Table 3 The relationship of risk factors and all-cause mortality between patients of survivor and non-survivor.

	OR (95% CI)	p-value	Adjusted OR
(95% CI)	p-value	
Age					
<65	Ref				
≥65	0.90 [0.39–2.09]	0.813			
Sex					
Female	Ref				
Male	0.74 [0.32–1.69]	0.473			
BMI					
<18.5	2.15 [0.65–7.05]	0.208			
18.5–24	Ref				
≥24	1.16 [0.47–2.88]	0.755			
Vaccinated times					
0	Ref				
≥1	0.63 [0.28–1.42]	0.264			
Etiology					
MAFLD	Ref				
Alcohol	2.10 [0.73–6.05]	0.168			
Virus	1.41 [0.52–3.85]	0.501			
Others	4.45 [0.752–6.26]	0.100			
Laboratory data					
Platelet (109/L)	1.00 [0.99–1.00]	0.820			
PT-INR	26.39 [3.65–190.63]	0.001	19.81 [1.31–299.49]	0.031	
Total bilirubin (mg/dL)	1.17 [0.81–1.68]	0.409			
ALT (U/L)	1.00 [0.99–1.00]	0.312			
AST (U/L)	1.00 [0.99–1.01]	0.191			
Albumin (g/dL)	0.09 [0.04–0.24]	<0.001	0.08 [0.02–0.39]	0.002	
hs-CRP (mg/L)	1.01 [1.01–1.02]	<0.001	1.02 [1.01–1.02]	0.001	
COVID-19–related liver injury					
ACLF or Decompensation	7.84 [2.80–21.97]	<0.001	0.92 [0.13–6.41]	0.930	
Acute liver injury	3.82 [1.61–9.06]	0.002	4.30 [0.83–22.37]	0.083	
Disease course and severity					
Length of hospital days	1.02 [1.00–1.04]	0.034	1.00 [0.97–1.03]	0.942	
ICU admission	2.59 [1.04–6.50]	0.042	0.90 [0.15–5.40]	0.909	
The use of vasopressors	7.54 [3.13–18.16]	<0.001	4.98 [1.27–19.46]	0.021	
The use of ventilator	3.12 [1.11–8.75]	0.031	0.45 [0.06–3.40]	0.439	
Notes:

Abbreviations: ACLF, acute-on-chronic liver failure; AST, aspartate aminotransferase; ALT, alanine aminotransferase; BMI, body mass index; CLD, chronic liver disease; ICU, intensive care unit; MAFLD, metabolic-associated fatty liver disease; hs-CRP, high-sensitivity C-reactive protein; OR, odds ratio; PT-INR, prothrombin time-international normalized ratio.

‡Calculated using logistic regression with the Firth approach by adjusting for albumin, hs-CRP, PT-INR, ACLF, ALI, the use of vasopressors, the use of a ventilator, ICU admission, length of hospital days, ALBI, FIB-4, and APRI.

Bold values indicate p < 0.05.

Analysis of liver indices and mortality between survivor and non-survivor groups

Using logistic regression, we analyzed the relationship between the ALBI score, FIB-4 index, and all-cause mortality, and the results are shown in Table 4. The multivariable analysis revealed significant differences in ALBI grade 3 (OR: 22.80, 95% CI [1.70–305.38], p = 0.018) and FIB-4 index ≥ 3.25 (OR: 10.62, 95% CI [1.12–100.31], p = 0.039), whereas no significant difference was observed in ALBI grade 2 (OR: 4.96, 95% CI [0.49–49.34], p = 0.174) and FIB-4 index between 1.45 to 3.25 (OR: 4.98, 95% CI [0.92–26.89], p = 0.062) between the survivor and non-survivor groups after adjusting for confounding factors.

Table 4 The relationship of ALBI score, FIB-4 index, and all-cause mortality between patients of survivor and non-survivor.

	OR (95% CI)	p-value	Adjusted OR (95% CI)	p-value	
ALBI score					
Grade 1	Ref		Ref		
Grade 2	5.56 [0.72–42.73]	0.100	4.96 [0.49–49.34]	0.174	
Grade 3	32.19 [3.55–291.56]	0.002	22.80 [1.70–305.38]	0.018	
FIB-4 index					
<1.45	Ref		Ref		
1.45–3.25	2.14 [0.78–5.89]	0.140	4.98 [0.92–26.89]	0.062	
≥3.25	5.56 [1.89–16.35]	0.002	10.62 [1.12–100.31]	0.039	
Notes:

Abbreviations: ALBI, albumin-bilirubin grade; FIB-4, fibrosis-4; OR, odds ratio.

‡Calculated using logistic regression with the Firth approach by adjusting for age, sex, BMI, vaccination, PT-INR, ALI, ALBI, FIB-4, APRI, ACLF, length of hospital days, ICU admission, the use of vasopressors, the use of ventilator.

Bold values indicate p < 0.05.

Discussion

To our knowledge, this multicenter cohort study is the first to emphasize hospitalized patients with non-cirrhotic CLD following COVID-19 during the omicron wave and investigate the association between liver fibrosis scores and mortality.

The ALBI score, FIB-4 index, and APRI are convenient and non-invasive methods for evaluating the liver status of patients with cirrhosis and non-cirrhotic CLD. Our results showed that among patients with non-cirrhotic CLD following COVID-19, only ALBI grade 3 (OR: 22.80, 95% CI [1.70–305.38], p = 0.018) and FIB-4 index ≥ 3.25 (OR: 10.62, 95% CI [1.12–100.31], p = 0.039) were independent risk factors for all-cause mortality, whereas ALBI grade 2 and FIB-4 index between 1.45 to 3.25 were not. According to consensus (Vallet-Pichard et al., 2007), a FIB-4 index ≥ 3.25 indicates fibrosis or pre-cirrhosis, equivalent to a METAVIR fibrosis score ≥ F3. FIB-4 index between 1.45 to 3.25 is roughly equivalent to a METAVIR fibrosis score F1–F2. Our results suggest that the ALBI score and FIB-4 index could be associated with mortality in the advanced stage (ALBI grade 3 and FIB-4 index ≥ 3.25) of non-cirrhotic CLD following COVID-19. In contrast, the mortality rate of patients with early-stage non-cirrhotic CLD (ALBI grades 1–2 and FIB-4 index < 3.25) did not increase significantly.

Several studies (Crisan et al., 2021; Kamal et al., 2022; Ao et al., 2022; Ibanez-Samaniego et al., 2020) have reported that the FIB-4 index can predict outcomes in patients with or without pre-existing CLD following COVID-19; however, none have examined the ALBI score for this issue. Bucci et al. (2022) analyzed 992 cirrhotic patients with COVID-19 and concluded that the FIB-4 index is a better predictor of mortality and severity than AST, ALT, and APRI. Li et al. (2021) enrolled the general population, including those with CLD (accounting for 32.2%), and revealed that the FIB-4 index was a predictor of mortality and severity in COVID-19. These results may be due to the predominant elevation of AST over ALT during COVID-19 in the context of systemic inflammation. Despite the widespread use of the FIB-4 index for outcome evaluation in previous studies, few studies have focused on patients with non-cirrhotic CLD. Only one small-sample registry study (Hatipoglu et al., 2023) assessed non-cirrhotic CLD and reported no significant change in the FIB-4 index among those who had COVID-19 compared to those who did not. However, unlike our study, they focused on changes in the FIB-4 value rather than stratifying the CLD stage.

Previous articles (Nagarajan et al., 2022; Kim et al., 2021a; Galiero et al., 2020; Kim et al., 2021b) have reported that CLD is associated with more severe COVID-19, worse outcomes, and higher mortality. However, most of these studies investigated the entire CLD cohort without separating non-cirrhotic CLD from cirrhosis. In our study of 3,087 patients, the overall in-hospital all-cause mortality rate was 10.92%, with 11.26% and 16.39% for non-cirrhotic CLD and cirrhosis, respectively. An early multinational cohort study including 745 patients reported a mortality rate of 32% in cirrhosis, with a stepwise trend by an upgrade of the Child-Pugh class (Marjot et al., 2021a). This study also indicated similar mortality between non-cirrhotic CLD and non-CLD patients, consistent with our results (11.26% vs. 10.89%). Widespread vaccination (Najjar-Debbiny et al., 2023) has decreased the omicron variant virulence (Wolter et al., 2022), and antiviral therapy is increasingly used (Najjar-Debbiny et al., 2023; Wong et al., 2022), resulting in the lowered mortality rate of COVID-19. Studies have revealed the effectiveness of antiviral therapy and vaccination in reducing severe COVID-19 and mortality during the omicron wave in patients in hospitalized (Najjar-Debbiny et al., 2023) and community settings (Wong et al., 2022). That can explain why the cirrhosis mortality rate in our study (16.39%) is lower than previous studies with non-omicron variants, ranging from 25–34% (Sarin et al., 2020; Kim et al., 2021a; Iavarone et al., 2020; Bajaj et al., 2021). However, in non-cirrhotic CLD, there was no difference in mortality rate in our study (11.26%) compared with non-omicron studies, ranging from 8–10% (Marjot et al., 2021a; Kim et al., 2021a). The severity and stage of liver disease seemed more relevant to all-cause mortality in patients with CLD following COVID-19.

In the current study, multivariable logistic regression was performed on non-cirrhotic patients with CLD to determine the risk factors for all-cause mortality. The results showed that PT-INR, albumin, hs-CRP, and vasopressor use were independent risk factors for mortality. In contrast, age, sex, BMI, vaccination, etiology, past medical history, platelet count, total bilirubin, ALT, AST, antiviral use, ventilator usage, hospitalization stay, and ICU admission were not significant risk factors. Additionally, we found that etiologies, including MAFLD, alcoholic, viral, and others, were not associated with mortality after adjusting for confounding factors. Previous studies (Marjot et al., 2021a; Kim et al., 2021a; Dufour et al., 2022; Mallet et al., 2021) have reported that only alcoholic liver disease (ALD), rather than MAFLD or viral infection, is a risk factor for mortality after COVID-19. In these studies, patients with ALD were predominantly included in the cirrhosis cohort rather than in the non-cirrhotic CLD cohort. ALD rarely presents in the early stage (Shah et al., 2019), for example, Marjot et al. (2021a) mentioned only 6% of patients with ALD in their cohort were not cirrhotic. In contrast, our study enrolled all non-cirrhotic CLD patients, relatively early stage of liver disease, which might explain the non-significance for mortality in ALD patients (OR: 2.10, 95% CI [0.73–6.05], p = 0.168). Furthermore, chronic viral hepatitis is prevalent in Taiwan, and our study found no significant association between viral etiology and mortality, consistent with previous studies (Marjot et al., 2021a). Therefore, our results support the inference that the severity and stage of pre-existing CLD determine the outcome rather than the etiology of CLD following COVID-19.

COVID-19-related liver injury is often subclinical, especially in healthy individuals, despite significant steatosis and fibrosis in some autopsies studies (Kaltschmidt et al., 2021; Lax et al., 2021). ALI accounts for 14–53% of patients with COVID-19 (Shiri Aghbash et al., 2022). The main manifestations of ALI are an increase in aminotransferases and bilirubin to varying degrees (Dufour et al., 2022; Chen et al., 2020). An increase in aminotransferases, particularly AST over ALT (Bernal-Monterde et al., 2020), is mainly attributed to the cytokine cascade, systemic inflammatory response, or drug-induced liver injury due to antivirals or tocilizumab (Shiri Aghbash et al., 2022), but less related to direct liver insult (Jothimani et al., 2020). The severity of systemic inflammation can be reflected by levels of hs-CRP, albumin, interleukin-6, and ferritin (Shiri Aghbash et al., 2022; Dufour et al., 2022; Effenberger et al., 2021; Da et al., 2021), which is consistent with our findings of elevated hs-CRP (OR: 1.02, 95% CI [1.01–1.02], p = 0.001) and decreased albumin levels (OR: 0.08, 95% CI [0.02–0.39], p = 0.002) in non-survivors compared to survivors. In addition, an increase in bilirubin levels, PT-INR, or the development of ALI or ACLF are markers of disease severity and poor prognosis in all CLD. However, in our cohort, only PT-INR was significantly elevated, possibly because early-stage CLD did not present with remarkable elevations in these markers.

Our study has several strengths. First, the ALBI score is a novel predictor of CLD, and to our knowledge, this is the first study to demonstrate that ALBI scores could be associated with the mortality of non-cirrhotic CLD following COVID-19. Second, we stratified the ALBI score and FIB-4 index based on the severity of non-cirrhotic CLD, which has rarely been discussed in previous studies. Our findings suggest that patients with ALBI grade 3 and FIB-4 index ≥ 3.25 upon admission are at significant mortality risk. We recommend that the ALBI grade and FIB-4 index be calculated for patients with non-cirrhotic CLD upon admission for COVID-19. Third, our study was conducted in Taiwan, where comprehensive health insurance and epidemic prevention policies minimize delays in seeking medical attention, which may result in a better representation of the actual circumstances of the disease. In addition, we enrolled patients with omicron variants; thus, our results may represent the recent COVID-19 situation.

However, this study has several limitations. First, this retrospective cohort study was conducted in a single country and ethnicity with small sample size, hospitalized patients, and excluded patients with missing data. Therefore, this may have resulted in selection bias, and the results may not be extrapolated to the entire population. Second, most patients had multiple underlying diseases, which may have confounded the risk factors for mortality. Therefore, we adjusted for confounding factors using multivariable logistic regression. Third, obtaining waist circumference and prediabetes information for patients with MAFLD from medical records was difficult, which may have caused underestimation in this group. Fourth, some indices and terms have various cut-off values or definitions, which may have resulted in different results than those of previous studies.

Conclusions

The early identification of high-risk individuals is critical for preventing mortality. Our study found that the severity and stage of pre-existing CLD determined the outcome more than the etiology following COVID-19. ALBI grade 3, FIB-4 index ≥ 3.25, higher PT-INR, hsCRP levels and lower albumin levels might be associated with poor clinical outcomes and disease severity.

Although more research is needed, we recommend that upon admission, clinicians should evaluate patients with non-cirrhotic CLD by checking their ALBI grade, FIB-4 index, PT-INR, hs-CRP, and albumin levels.

Supplemental Information

Supplemental Information 1 Raw data.

*Abbreviations: AST, aspartate aminotransferase; ALT, alanine aminotransferase; ALBI, albumin-bilirubin grade; APRI, aspartate aminotransferase to platelet ratio index; BMI, body mass index; COPD, Chronic Obstructive Pulmonary Disease; FIB-4, fibrosis-4; HCC, hepatocellular carcinoma; ICU, intensive care unit; K, potussium; MAFLD, metabolic associated fatty liver disease; hs-CRP, high-sensitivity C-reactive protein; Na, sodium; OR, odds ratio; PT-INR, prothrombin time-international normalized ratio.

Click here for additional data file.

We would like to express our sincere gratitude to the statistician, Yu-Cih Wu, who conducted the data analysis for this study. We thank her for her valuable insights and advice on the statistical methods used in this study.

Additional Information and Declarations

Competing Interests

Author Contributions

Ethics

Data Availability

The authors declare that they have no competing interests.

Pei-Jui Wu performed the experiments, analyzed the data, authored or reviewed drafts of the article, and approved the final draft.

I-Che Feng analyzed the data, prepared figures and/or tables, and approved the final draft.

Chih-Cheng Lai conceived and designed the experiments, analyzed the data, authored or reviewed drafts of the article, and approved the final draft.

Chung-Han Ho performed the experiments, prepared figures and/or tables, and approved the final draft.

Wei-Chih Kan performed the experiments, prepared figures and/or tables, and approved the final draft.

Ming-Jen Sheu conceived and designed the experiments, prepared figures and/or tables, and approved the final draft.

Hsing-Tao Kuo conceived and designed the experiments, authored or reviewed drafts of the article, and approved the final draft.

The following information was supplied relating to ethical approvals (i.e., approving body and any reference numbers):

The study was approved by the Institutional Review Board of Chimei Medical Center. The informed consent requirement was waived due to the study’s retrospective nature.

The following information was supplied regarding data availability:

The raw data are available in the Supplemental File.

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
