# Peer review of "The mortality of hospitalized patients with COVID-19 and non-cirrhotic chronic liver disease: a retrospective multi-center study"

_PeerJ, doi:10.7717/peerj.16582_

## Round 0.1 · original submission · Major Revisions

In addition to the reviewer's comments, please see my comments below:
1. You carried out a multivariable analysis, not a multivariate
2. More details on the selection of variables for multivariable logistic regression are needed.
3. I have the following recommendations for improving the validity of the findings, including adding excluded patient numbers, clarifying variable usage, and justifying the inclusion of certain factors in regression analysis.

Reviewer 1 ·

Basic reporting

no comment

Experimental design

no comment

Validity of the findings

no comment

Additional comments

Older patients with chronic diseases have a higher risk of severity and mortality when infected with COVID-19. These patients need more attention and evaluation upon admission. According to the inclusion and exclusion criteria, this retrospectively study ultimately focused on 231 patients with non-cirrhotic CLD and COVID-19 in three hospitals. It found that non-survivors had higher levels of PT-INR, ALT, AST and hs-CRP and lower albumin levels. ALBI grade 3 and FIB-4 index ≥3.25 were proved to be useful predictors of mortality in non-cirrhotic CLD patients with COVID-19. The conclusions may be valuable to predict severity and mortality of Omicron-infected patients with non-cirrhotic CLD. It also reminds physicians for early intervention to minimize mortality in critical patients. The writing is logical, fluent and readable. However, there are a few concerns or context errors which should be addressed and revised before publication.
1. According to the described results, in “Discussion” part line199-201, ALBI grade 3 and FIB-4 index ≥3.25 could predict mortality, not survival in non-cirrhotic CLD patients with COVID-19;
2. As described in the “Results”, The mortality rate of all 3,087 patients was 10.92% (n = 337). The all-cause mortality rates of patients with non-cirrhotic CLD and cirrhosis were 11.26% and 16.39%, respectively. In “Discussion” part line 224-226, 10.92% is not the mortality value of non-CLD patients in this study. It should be corrected or recalculated.

Reviewer 2 ·

Basic reporting

no comment

Experimental design

1. The word “Predictors” in the title indicates that the article aims to predict mortality in NCLD patients with COVID-19, while the method and results are the risk factors of the mortality of the target objective because there is no evaluation of the predictive capacity of the “predictor”. A high OR does however not directly mean that a marker has a high additional value, since a positive marker value may be quite rare. (Steyerberg EW, Pencina MJ, Lingsma HF, Kattan MW, Vickers AJ, Van Calster B. Assessing the incremental value of diagnostic and prognostic markers: a review and illustration. Eur J Clin Invest. 2012 Feb;42(2):216-28. doi: 10.1111/j.1365-2362.2011.02562.x. Epub 2011 Jul 5. PMID: 21726217; PMCID: PMC3587963.)
2. Definitions :(1) Line 110: The description of ‘cirrhosis is the end stage of liver disease’ is inaccurate.
(2) Line 107: Could the diagnosis of CLD be more specified?
(3) Line 119: What’s the duration of elevated ALT/AST in ALI? Could ALI be considered as a form of CLD?
(4) Line 121: Which guideline does the ACLF definition refer to? Please add a reference and check that the definition is accurate.
(5) Definitions and grading of some variables such as FIB-4 and ALBI score should be presented.
3. Statistical analysis: Please elaborate on the selection of variables to be included in the multivariate logistic regression.

Validity of the findings

1. Add the excluded patient’s number to each exclusion reason in the first exclusion box in the enrollment flow diagram.
2. In the notes of Table 3, is “the use of dexamethasone” redundant?
3. Are ALBI score and FIB-4 included in multivariate logistic regression without the laboratory data(except PT-INR), COVID-19-related liver injury(except ALI), and disease course and severity? If so, why?

Additional comments

no comment.

---

## Round 0.2 · Major Revisions

One of the reviewers has raised critical issues with your submission. I will reconsider your manuscript if those issues are critically and adequately addressed.

Reviewer 1 ·

Basic reporting

No comments.

Experimental design

No comments.

Validity of the findings

No comments.

Additional comments

In a short period, the author replied to the editor’s and reviewers’ comments in details. A flow diagram of patient enrollment and several professional definitions were added for better understanding. However, there are a few more concerns or context errors which should be deliberated and revised before next submission.
1.Although the title has been revised to “The mortality of hospitalized patients with COVID-19 and noncirrhotic non-cirrhotic chronic liver disease”, the abstract and the whole contents are discussing the predictors of mortality in hospitalized non-cirrhotic chronic liver disease following COVID-19. According to findings in this article, the so-called predictors and the mortality present correlation rather than prediction or causality. Maybe it’s more appropriate to conclude that ALBI grade 3, FIB-4 index ≥ 3.25, higher PT-INR, hsCRP levels and lower albumin levels contribute to the mortality in non-cirrhotic CLD patients with COVID-19. If author aims to explore the predictors of the mortality of patients with COVID-19 and non-cirrhotic chronic liver disease, more indexes should be considered for inclusion in the study, like COVID-19 severity grades, pneumonia lesion, oxygen therapy method and duration, renal function and lactate level, etc. The cut-off values of each index should be supplemented.
2.As described in “Discussion” Line 239, COVID-19-related pneumonia was the leading cause of mortality (92.31%). From here we see that, the liver status and fibrosis indices are not the most important lethal factors. Is there a relationship between the liver status, fibrosis indices and COVID-19 severity? Perhaps this is a breakthrough point. Also, it’s better to display and analyze the classification and proportion of various causes of death.
3.As seen in Figure 1, there are 152 patients who were excluded for lack of available clinical data. This is a great pity.
4.In “Results - Clinical characteristics” part line 157, 385 (exactly 324) were excluded based on the exclusion criteria. It should be corrected.

Reviewer 2 ·

Basic reporting

no comment

Experimental design

no comment

Validity of the findings

no comment

Additional comments

no comment

---

## Round 0.3 · accepted · Accept

All comments have been adequately addressed.